# Scrutinizing Relationships between Submarine Groundwater Discharge and Upstream Areas Using Thermal Remote Sensing: A Case Study in the Northern Persian Gulf

Aliakbar Nazari Samani [1], Mohsen Farzin [2], Omid Rahmati [3], Sadat Feiznia [1], Gholam Abbas Kazemi [4], Giles Foody [5] and Assefa M. Melesse [6,*]

1 Department of Reclamation of Arid and Mountainous Regions, Faculty of Natural Resources, University of Tehran, Karaj 31587-77871, Iran; aknazari@ut.ac.ir (A.N.S.); sfeiz@ut.ac.ir (S.F.)
2 Department of Range and Watershed Management, Faculty of Agriculture and Natural Resources, Yasouj University, Yasouj 75918-74934, Iran; m.farzin@yu.ac.ir
3 Soil Conservation and Watershed Management Research Department, Kurdistan Agricultural and Natural Resources Research and Education Center, AREEO, Sanandaj 66169-36311, Iran; o.rahmati@areeo.ac.ir
4 Department of Hydrogeology, Faculty of Earth Sciences, Shahrood University of Technology, Shahrood 36199-95161, Iran; g_a_kazemi@shahroodut.ac.ir
5 School of Geography, University of Nottingham, Nottingham NG7 2RD, UK; giles.foody@nottingham.ac.uk
6 Department of Earth and Environment, Institute of Environment, Florida International University, Miami, FL 33199, USA
* Correspondence: melessea@fiu.edu

**Abstract:** Nutrient input through submarine groundwater discharge (SGD) often plays a significant role in primary productivity and nutrient cycling in the coastal areas. Understanding relationships between SGD and topo-hydrological and geo-environmental characteristics of upstream zones is essential for sustainable development in these areas. However, these important relationships have not yet been completely explored using data-mining approaches, especially in arid and semi-arid coastal lands. Here, Landsat 8 thermal sensor data were used to identify potential sites of SGD at a regional scale. Relationships between the remotely-sensed sea surface temperature (SST) patterns and geo-environmental variables of upland watersheds were analyzed using logistic regression model for the first time. The accuracy of the predictions was evaluated using the area under the receiver operating characteristic curve (AUC-ROC) metric. A highly accurate model, with the AUC-ROC of 96.6%, was generated. Moreover, the results indicated that the percentage of karstic lithological formation and topographic wetness index were key variables influencing SGD phenomenon and spatial distribution in the northern coastal areas of the Persian Gulf. The adopted methodology and applied metrics can be transferred to other coastal regions as a rapid assessment procedure for SGD site detection. Moreover, the results can help planners and decision-makers to develop efficient environmental management strategies and the design of comprehensive sustainable development policies.

**Keywords:** thermal remote sensing; submarine groundwater discharge; geo-environmental variables; Persian Gulf; karstic formations

## 1. Introduction

Submarine Groundwater Discharge (SGD) is defined as any water subsurface flow from the land into the sea. Recognizing the area having this flow is very important for hydrological and ecological studies. SGD is an important pathway from the terrestrial to the marine environment that plays a significant role in hydrological and ecological processes such as: nutrient cycling, geochemical mass balances, and primary productivity in the coastal waters [1,2]. The importance of SGD as a source of nutrients, carbon and trace metals to coastal waters in water resources management and marine ecology has become increasingly recognized [3–6]. SGD has important impacts on variables such as water

quality and phytoplankton dynamics which in turn relate to issue such as algal blooms and eutrophication [7]. Moosdorf and Oehler [4] indicate that SGD resources have five major application areas: water for drinking, agriculture, hygiene, fishing/diving, and spiritual use. Consequently, there is a demand for SGD water and also interest in its quality. For example, the governor of Florida, USA, proposed a plan to transfer water from one of the largest submarine spring system in state (Spring creek) to Miami to help meet its freshwater supply needs [8]. However, wastewater injection, fertilized agricultural lands, and areas with high septic-cesspool system density have the potential to contribute excess nutrients to coastal waters via the SGD which can lead to environmental deterioration of coastal zones [9,10]. Additional concerns may be present in some regions. For example, Garcia-Orellana et al. [11] reported that SGD can increase the natural radioactivity levels in coastal lagoons. Therefore, sustainable management of coastal waters requires a comprehensive assessment of the relationship between SGD and geo-environmental variable such as geology, topography.

Over the last decade, numerous studies worldwide have successfully applied radon and radium isotopes to quantify SGD fluxes over a range of different time-scales, estimate the magnitude of SGD and determine its relative importance in chemical budgets of coastal waters [12–15]. However, the behavior of radium and radon in coastal aquifers is complex [16], and also laboratory experiments of radioisotopes are impossible in developing countries. Although hydrogeological modeling and isotope-based approaches have some limitations for analyzing relations between geo-environmental variables and SGD at regional scales and especially are extremely costly and time-consuming, the proposed methodology overcomes the difficulties and time required for field surveying. Other methods for mapping SGD such as ground electrical resistivity surveys are only suited for use over small areas ($\sim$100 m$^2$) [17]. Alternative methods for studying the SGD at local to regional scales are required.

Among the techniques employed to assess SGD, thermal infrared (TIR) remote sensing using satellite or airborne sensors can be applied to explore groundwater discharge sites along a shoreline [18]. Normally groundwater tends to occur at the average annual temperature of groundwater and, therefore, can be thermally distinct from surface-waters [19]. Identification of SGD using TIR remote sensing is possible in areas where there is significant thermal contrast between the receiving surface-water body and the discharging pore fluid [20,21]. Indeed, remote sensing-based methods are not only useful in understanding SGD patterns in coastal environments, but also help in determining geological heterogeneity at a relatively high spatial resolution and over large areas [22]. The potential of TIR remote sensing has been explored in various regions around the world [20,23]. Importantly, satellite TIR remote sensing has been found to be an effective tool for detecting SGD. For example, Wilson and Rocha [24] used time-series Landsat TIR data (medium resolution satellite imagery) to identify over 30 new sources of SGD along the fractured bedrock coast of Ireland. Sass et al. [25] detected terrestrial groundwater discharge zones with Landsat TIR data from Alberta, Canada. Arricibita et al. [26], who used a TIR camera in a laboratory experiment, indicated that analysis of TIR data allows for the measurement of water surface temperature at high spatial resolution across a wide range of scales. Thus, TIR remote sensing can be applied to assess SGD and extrapolate local groundwater fluxes to a regional scale and, therefore, potentially reduce the amount of field sampling and in situ measurements required.

One region where SGD has not been investigated in detail is along the Persian Gulf coastline [27], despite the presence of several well-known karstic springs and its important aquatic ecosystems. Additionally, the impact of geo-environmental variables of the local upland area (e.g., topography, geomorphometric, vegetation cover, geology) on SGD occurrence has not been investigated. In policy terms, a need was identified by Iranian Department of Water Resources Management and Iran National Science Foundation (INSF) in this region, to investigate SGD along the Persian Gulf coastline. Thus, this study aims to develop an integrated framework which applies remote sensing and statistical analyses to

develop and improve tools for providing useful information on the recognition of potential sites of SGD. The research was supported by geochemical measurements and in situ field measurements of water temperature. Statistical methods have not been widely used to model SGD despite their considerable potential. In particular, logistic regression analysis, which has been used in a range of environmental science applications [28–32], may be well-suited for SGD modelling. In a logistic regression, the dependent variable is binary or categorical, whereas its independent variables could be a mixture of continuous and binary or categorical variables. In addition, the assumption of normality is not needed for logistic regression. According to these key features, logistic regression is advantageous to model the probability of SGD compared to other statistical methods like simple regression. The specific objectives of this study are to (1) explore the TIR response of coastal waters along the Persian Gulf and detect locations of SGD, (2) establish statistical relationships between a SGD (dependent variable) and a set of spatial predictors of the upland area, and (3) evaluate the capability and robustness of proposed method using in situ measurements.

## 2. Material and Methods

### 2.1. Study Area

The Persian Gulf is a semi-enclosed marine system surrounded by eight countries, and it is located to the south of Iran (Figure 1). It has a total area of approximately 240,000 km², making it one of the largest gulf areas in the world, and also known as a major center for the oil industry [33]. The Persian Gulf is a shallow sea which characterized by warm and saline water. Its depth generally increases from west to east with a maximum depth of 90 m in the Strait of Hormuz and an average depth of 36 m. The average tidal range in this region is 1–1.5 m. Although there is a high evaporation rate in the Persian Gulf, the water loss is compensated by a surface current moving counter clockwise from the Indian Ocean to the Oman Sea and Persian Gulf [34].

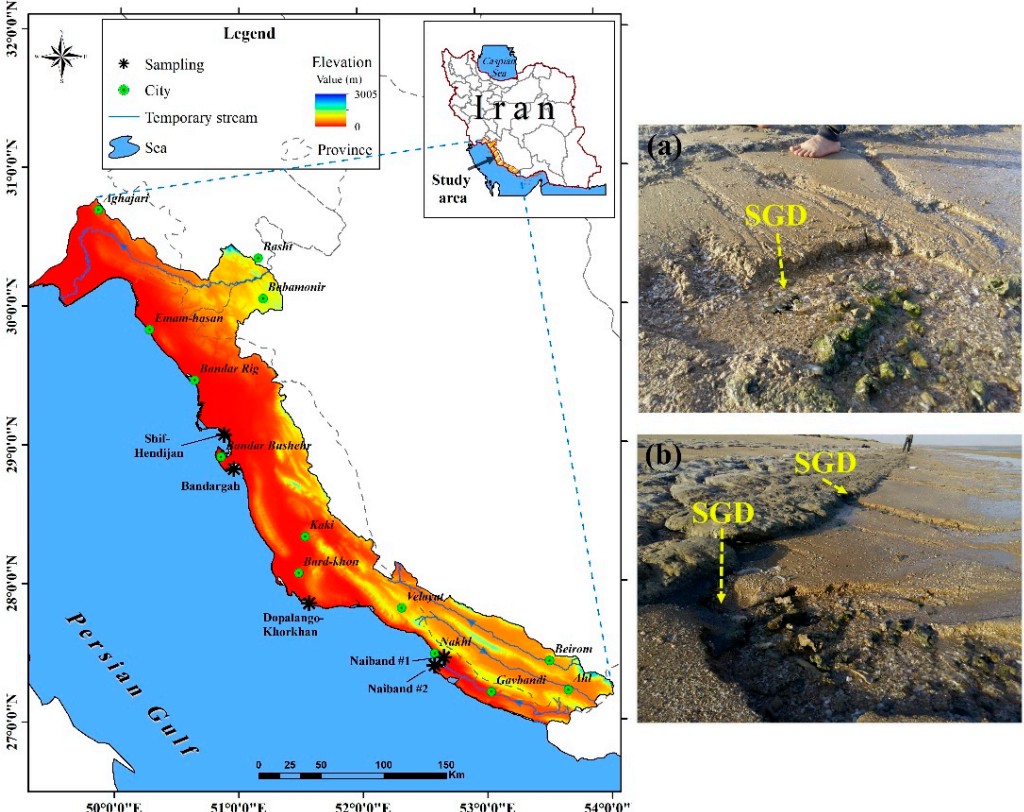

**Figure 1.** Location map of the study area in the south of Iran. Field photographs of some SGD occurrences in the study area: (**a**) Shif-Hendijan, and (**b**) Naiband #1.

The northern part of the Persian Gulf is the study area and generally described as a very shallow water with a mean depth of 5 m. The study area comprises some karstic coastal aquifers and submarine groundwater springs near the coastal zone. The study area located in five provinces: Boushehr, Zkhozestan, Hormozgan, Fars, and Kohgiluyeh-Boyer-Ahmad. From a hydrological viewpoint, there are some temporary streams in the study area that are dry in summers. Water quality is substantially influenced by various industrial and agricultural outputs, discharging their wastewater directly to the sea or via temporary streams. This coastal region has also experienced rapid urban and industrial development as well as touristic growth over the last decade, leading to increased demand for water consumption. In addition, the increasing array of anthropogenic interferences have substantial negative impacts on marine ecosystems. There is, therefore, a desire to study, SGD potential and the variables that influence this important resource in the region [35].

### 2.2. Methodology

The methodology is summarized in Figure 2 and has four main parts:

i.    Formation of sea surface temperature (SST) and standardized temperature anomaly (STA) maps from TIR imagery.
ii.   Identification of thermal anomalies as potential sites of SGD.
iii.  Selection of geo-environmental variables
iv.   Spatial analysis and using three different buffer zones
v.    Modeling the relationships between SGD and geo-environmental characteristics of upstream zones.
vi.   Assessing the accuracy of the model and undertaking a sensitivity analysis.

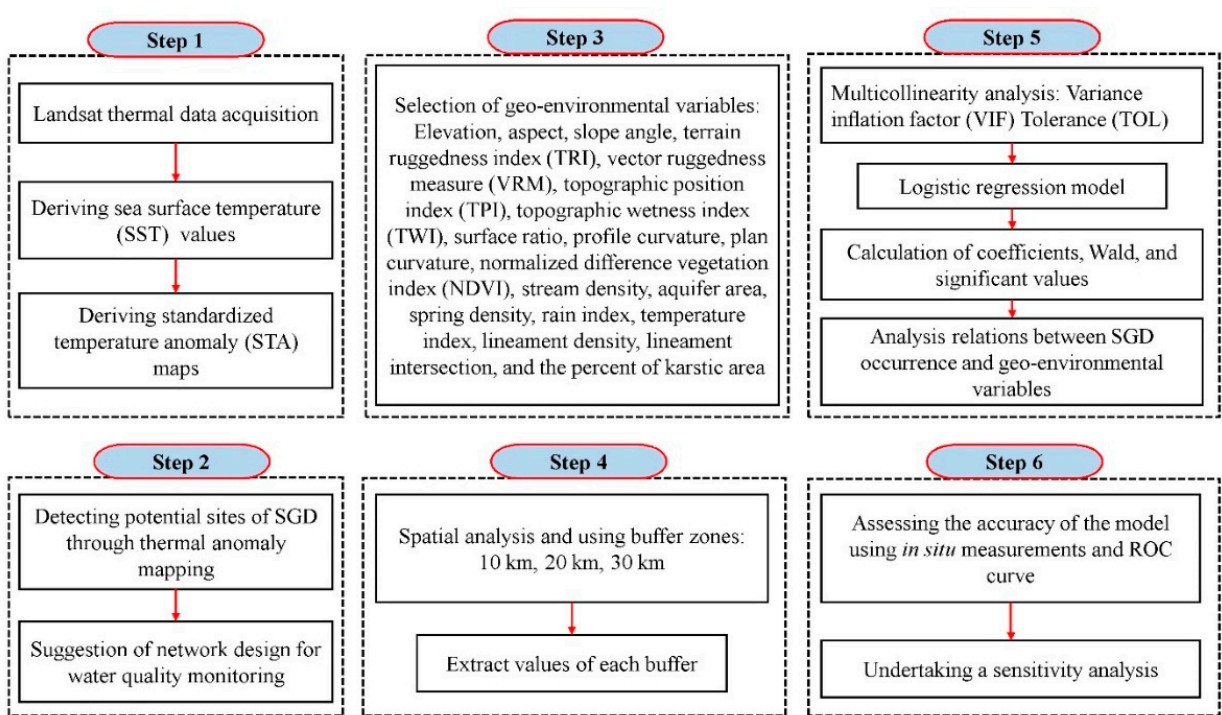

**Figure 2.** Flowchart of the study.

### 2.2.1. Landsat Thermal Data Acquisition

Although fine spatial resolution airborne [36], ground-based thermal imaging systems [37], and handheld thermal sensors are effective, these systems tend to be extremely costly and unsuitable for application to very large areas especially if continued monitoring of groundwater discharges is desired. Hence, thermal infrared images acquired by the thermal infrared sensor (TIRS) carried on the Landsat-8 satellite with a spatial resolution

of 100 m were used. Landsat offers a potential of 16-day revisit capability. Following the literature [24], the imagery was acquired during the late spring and summer (from May to September) when the maximum temperature differences between surface water and groundwater occurs in the Persian Gulf. Attention was focused on the Persian Gulf itself and to aid the analysis, land was excluded through the use of a land-sea mask that had been generated from an earlier Landsat ETM+ near-infrared image. Fortunately, in this period satellite images can be obtained in cloud-free days. Ten images were acquired for 2015 and 2016 (Row/Pass: 162/41, 163/40, 163/41, 164/39, 164/40). The Row denotes to the latitudinal center line of a frame of imagery while the Path refers a line that the satellite moves along it. The combination of a Path number and a Row number uniquely identifies a nominal scene center. All of the images obtained were cloud free. The Landsat-8 has an equatorial crossing time at 10:00 a.m. +/− 15 min (local time). Therefore, the time, when Landsat-8 crosses the Persian Gulf, was close to 10:00 a.m. local time. Fortunately, the Persian Gulf maximum temperature differences between groundwater and surface water exist in this time period and thermal anomalies can be detected using satellite remote sensing. In the next step, land pixels in each image were masked.

### 2.2.2. Thermal Infrared Image Processing

To identify thermal anomalies, an automated thermal anomaly extraction technique based on a moving window was used [38]. As an initial step, pixel digital numbers (DNs) of the Landsat TIR band 10 were converted to top-of-atmosphere (*TOA*) spectral radiance using Equation (1) [39]:

$$L_{\lambda TOA} = M_L Q_{Cal} + A_L \tag{1}$$

where, $L_{\lambda TOA}$ is *TOA* spectral radiance (Watts $(m^2 \cdot sr \cdot \mu m)^{-1}$), $M_L$ is rescaling factor ($3.342 \times 10^{-4}$ for Landsat-8 band 10), $Q_{cal}$ is DN values, and $A_L$ is rescaling factor (0.1 for Landsat-8 band 10) [40].

The *TOA* values were corrected for atmospheric effects (Equation (2)) to determine surface water radiance using parameters derived from NASA's online atmospheric correction tool [24]. The atmospheric correction was applied to prevent changes due to atmospheric effects being interpreted as changes in the water body. Atmospheric correction parameters derived from an online atmospheric correction parameter tool (http://atmcorr.gsfc.nasa.gov/) were used to derive scene at-surface kinetic sea temperature values.

$$L_{\lambda T} = \frac{L_{\lambda TOA} - L_{\lambda UP}}{\tau \epsilon} - \frac{1 - \epsilon}{\epsilon}(L_{\lambda DOWN}) \tag{2}$$

where, $L_{\lambda T}$ is the radiance of a blackbody target of kinetic temperature $T$ ($Wm^{-2} sr^{-1} \mu m^{-1}$), $\tau$ is the atmospheric transmission (unitless), and $\varepsilon$ is emissivity of water (ranges from 0.98 to 0.99). $L_{\lambda TOA}$ is calculated from Equation (1). In this study, a constant emissivity of 0.989 was used as suggested in the literature [40]. $L_{\lambda UP}$ and $L_{\lambda DOWN}$ are upwelling (atmospheric path radiance) and downwelling (sky radiance), respectively. Finally, surface water radiance values were converted into temperature (Equation (3)) [41]:

$$T_{ss} = \frac{K_2}{ln\left(\frac{K_1}{L_{\lambda T}}\right) + 1} \tag{3}$$

where, $T_{ss}$ is the sea surface temperature (SST) (Kelvin). $K_1$ and $K_2$ are band-specific thermal conversion constants obtained from the available metadata [42].

### 2.2.3. Assessment of Thermal Anomalies

Heat has been considered as a groundwater tracer for over a century and remote sensing-based methods for SGD detection are appropriate where temperature gradients form between discharging groundwater and the surface water bodies [19]. The use of Landsat TIR data to detect thermal anomalies has been successfully demonstrated in previous studies e.g., [43], and these may be used to assess the spatial distribution of SGD. In winter months the SGD will be warmer than the receiving surface-waters but in

summer SGD will be cooler than surface-waters [18,44]. To determine the geographical location of potential sites of SGD, a set of temperature anomaly (*TA*) and standardized temperature anomaly (*STA*) maps was generated from each of the SST layers produced from the remotely sensed imagery. *TA* has been defined as the difference between the SST value of each pixel and the average SST value estimated for the coastal water body (Equation (4)) [24]:

$$TA = T_p - T_a \qquad (4)$$

where, *TA* is temperature anomaly (Kelvin), $T_p$ denotes the temperature value specific to each pixel in the scene (Kelvin), and $T_a$ is the average temperature value for the scene (Kelvin). STA (dimensionless) can be calculated using the following equation (Equation (5)) [24]:

$$STA = \frac{TA}{\sigma} \qquad (5)$$

where, $\sigma$ is the standard deviation of *SST* values.

According to obtained thermal anomalies, SGD and non-SGD (without submarine groundwater discharge process) locations were identified. The frequency of thermal anomalies in both 2015 and 2016 was considered as a criterion for calculating areas of thermal anomaly. To confirm this classification, comparison was made with the temperature of water samples that were obtained at five sites: Naiband #1, Naiband #2, Dopalango-Khorkhan, Bandargah, and Shif- Hendijan (Figure 1). At each site, four water samples were collected (*n* = 20).

### 2.2.4. Statistical Modeling
#### Dependent and Independent Variables

There are no universal guidelines for selecting independent variables that influence SGD. Here, several geo-environmental variables including geological, environmental and topo-hydrological variables were selected to evaluate the relationship between SGD occurrence and upstream characteristics. These variables were: elevation (m), slope angle (%), aspect, terrain ruggedness index (TRI) (m), vector ruggedness measure (VRM), topographic position index (TPI) (m), topographic wetness index (TWI), surface ratio, profile curvature (Radians m$^{-1}$), plan curvature, normalized difference vegetation index (NDVI), stream density (Km km$^{-2}$), aquifer area (Km$^{-2}$), spring density, annual precipitation (mm), air temperature (°C), lineament density (Km km$^{-2)}$, lineament intersection, and the percent of karstic area (PKA). The calculations of these geomorphometric and topo-hydrological variables have been widely reported in the literature [44,45]. SGD occurrences were considered as the dependent variable in the analyses. All parameters had a scale of 1:50,000 except percent of karstic area SGD sites, which had a scale of 1:100,000. Furthermore, all variables had a grid GIS data type except SGD sites, which was a polygon GIS data type.

A variety of data sources were used to obtain data on the independent variables. A digital elevation model (DEM) with pixel size of 20 m was generated from 1:50,000-scale topographic maps of the study area. The altitude, slope angle, aspect, TRI, VRM, TPI, TWI, surface ratio, profile curvature, and plan curvature were produced based on the DEM using SAGA-GIS software (System for Automated Geoscientific Analyses). The NDVI was calculated from the red and near infrared (NIR) Landsat 8 OLI bands to show land cover situation. Stream density layer was generated using the existing stream network of study area. Spring density was also produced in ArcGIS software using available spring inventory map—obtained from Iranian Department of Water Resources Management. All lineaments were extracted from a mosaic of Landsat images using edge enhancement and filtering techniques as well as subsequent field verifications. Then, lineament density and lineament intersection layers were produced in ArcGIS 10.2 software. Geological maps at 1:100,000-scale covering the study area were obtained from Geological Survey Organization and different geological units were identified. Lithological groups and faults were extracted from these available geological maps. As a pre-process step, all layers were resampled to the coarsest resolution data set of 1:100,000 before analysis. All of above-mentioned

variables were extracted for upland coastal area and therefore three different buffer zones including 10, 20, and 30 km were built from each SGD location to the upland regions using ArcGIS 10.3. In this study, buffer zones were selected based on the spatial scale of the study as well as the distance of SGD sites from upland areas. Finally, all raster values of each variable were extracted by each buffer polygon for both SGD and non-SGD sites.

Logistic Regression Analysis

Logistic regression (LR) has been widely used in analyzing geohazards and a range of other earth science applications [46]. Its goal is to find the best fitting model to describe the relationship between dependent variable (the presence or absence of SGD) and a set of independent variables (geo-environmental variables). An advantage of the LR model is that dependent variable could be binary or categorical and the independent variables may be either continuous or categorical and they do not necessarily have to follow a normal distribution [47]. Maximum likelihood estimation is applied after transforming the dependent variable into a logit variable which allows the estimation of the probability of a certain event occurring [48]. The LR model establishes a functional relationship between the binary coded SGD locations (absence or presence of a SGD) and different variables that are recognized as playing a role in SGD and hydrogeologic processes. Further details on the LR model can be found in Hosmer et al. [49] and Kleinbaum and Klein [50] but the general form of LR model is as follows:

$$P = \frac{1}{1 + e^{-z}} \tag{6}$$

where $P$ is the probability of an event (SGD) occurrence, which varies from 0 to 1 on an s-shaped curve. In addition, the parameter $z$ can be calculated with the following equation:

$$z = b_0 + b_1 x_1 + b_2 x_2 + \cdots + b_n x_n \tag{7}$$

where, $b_0$ is the intercept of the model, $b_i$ ($i = 1, 2, 3, \ldots, n$) is the slope coefficient of the model, $x_i$ ($i = 1, 2, 3, \ldots, n$) is the independent variable, and $n$ is the number of independent variables. If $z$ is denoted as a binary response variable (0 or 1), value 0 ($z = 0$) indicates the absence of a SGD (non-SGD location) and value 1 ($z = 1$) means the presence of a SGD. One of the main advantages of this type of analysis is that the relative importance of the independent variables (i.e., contribution in modeling) can be determined using the coefficients of the regression function. As mentioned above, all of the independent variables were extracted from the upland area defined by the three buffer areas used.

Model fitting using LR is sensitive to collinearities among the independent variables. The variance inflation factor (VIF) and Tolerance (TOL) are two important parameters for the identification of multicollinearity [51]. TOL smaller than 0.1 suggests serious multicollinearity and also TOL $\geq$ 10 is an indicator for multicollinearity between independent variables [50]. The TOL and VIF values in this study shows no serious multicollinearity between the independent variables (predictors). The pseudo $R^2$ value in LR analysis cautiously indicates how the logit model fits the dataset and can be computed from $1 - (ln \text{ likelihood}_{\text{finalstep}} / ln \text{ likelihood}_{\text{initial}})$ [52]. Thus, a pseudo $R^2$ equal to 1 shows a perfect fit, whereas 0 indicates no relationship [51].

Validation and Sensitivity Analysis

Salinity cannot be considered as a reliable criterion to compare SGD and non-SGD sites because SGD includes both recirculated submarine groundwater discharge caused by recirculation of intruded seawater and fresh submarine groundwater discharge induced by hydraulic head difference between inland groundwater and seawater. Therefore, in situ field measurements of water temperature was used to verify potential sites of SGD in 2015 and 2016. The accuracy of the model was evaluated using the receiver operating characteristics (ROC) curve. The ROC plot shows the true positive rate (TPR) as a function

of false positive rate (FPR). The TPR and FPR can be obtained based on a confusion matrix using the following equations:

$$TPR = \frac{TP}{TP + FN} \tag{8}$$

$$FPR = \frac{FP}{FP + TN} \tag{9}$$

where *TN* (true negative) and *TP* (true positive) are the number of pixels that are correctly classified as SGD or non-SGD whereas *FN* (false negative) and *FP* (false positive) are the numbers of pixels erroneously classified. The area under the ROC curve (AUC-ROC) was considered as a threshold-independent evaluation criterion [52,53]. The SGD inventory map was randomly split into two groups: (i) the training dataset, which comprised 70% of the SGD inventory used in the training/calibration phase of the model; and (ii) the validation dataset, which contained the remaining 30% of the inventory.

To perform sensitivity analysis, the relative decrease (RD) of AUC-ROC values was also considered [54]. Sensitivity analysis allows the investigation of the dependency of the model output on the influence of the conditioning variables. It is the decrease in AUC when the variable is removed from the model. The *RD* can be calculated from Equation (10).

$$RD = \frac{([AUC - ROC]_{all} - [AUC - ROC]_i)}{[AUC - ROC]_{all}} \times 100 \tag{10}$$

where $AUC - ROC_{all}$ and $AUC - ROC_i$ indicate the $AUC - ROC$ values obtained from the SGD prediction using all independent variables and the prediction when the $i_{th}$ independent variable has been excluded, respectively.

## 3. Results

### 3.1. Temperature and Thermal Anomaly Mapping

Figure 3 shows one of the SST maps derived from 60 m resolution Landsat ETM+ TIR images acquired on 23 August 2015. SST in this map ranged from a minimum of ~24 °C to a maximum of ~39 °C. Clearly discernible cold-water plumes and potential SGD locations emanating from some nearshore waters along the coastline.

To facilitate a context-based inter-comparison of temperature anomalies, a STA map can reveal the relative significance of the anomalies observed at different locations. An example of STA from August 23 is shown in Figure 4. In this map, cold water plumes are evident and can be interpreted to delineate the location and extent of groundwater discharge—negative values indicate pixels associated with SGD. In all STA maps produced during 2015–2016, STA values range from –5.73 to 23.81. In order to facilitate the interpretation of STA map and delineation of the groundwater discharge, each STA map was reclassified (Figure 5). The largest negative STA values were detected within plumes mapped off the coastline south of Kangan, Bandargah, Bandar Rig, Hendijan, east and west of Bandar Boushehr, Naiband, Dopalango, and Khorkhan. Visual inspection of the processed Landsat scenes revealed potential SGD sites in the northern part of the Persian Gulf could highlight new SGD sites that had previously unidentified links between aquifers on land and gulf. From a hydrogeological viewpoint, these potential sites are generally characterized by a faulted, fractured and permeable bedrock geology comprising predominantly limestone, sandstone or mudstone associated with locally productive aquifer types and highly conducive to the transmission of water. In addition, according to the geological surveys, it is apparent that the presence of karst structures, bedrock fissures and faults adjacent to the thermal plumes is serving as a hydrogeological pathway transporting potentially large volumes of groundwater and associated materials to the sea.

Figure 6 indicated the geometric intersection of the anomalies. According to this figure, evidently discernible cold-water plumes emanate from nearshore waters along Naiband, Asaloye, Dopalango, Dahane Tahmadan, Khorkhan, and Bandar Busher coastlines. This map is considered as the result of the remote sensing analysis in this study and subsequently is used in the statistical modeling.

The STA maps generated were overlaid in a GIS environment. Table 1 shows the area of thermal anomalies (i.e., negative values in STA maps) in 2015 and 2016 and their overlapping surface area in this time period. Anomalous areas in common may highlight locations of SGD.

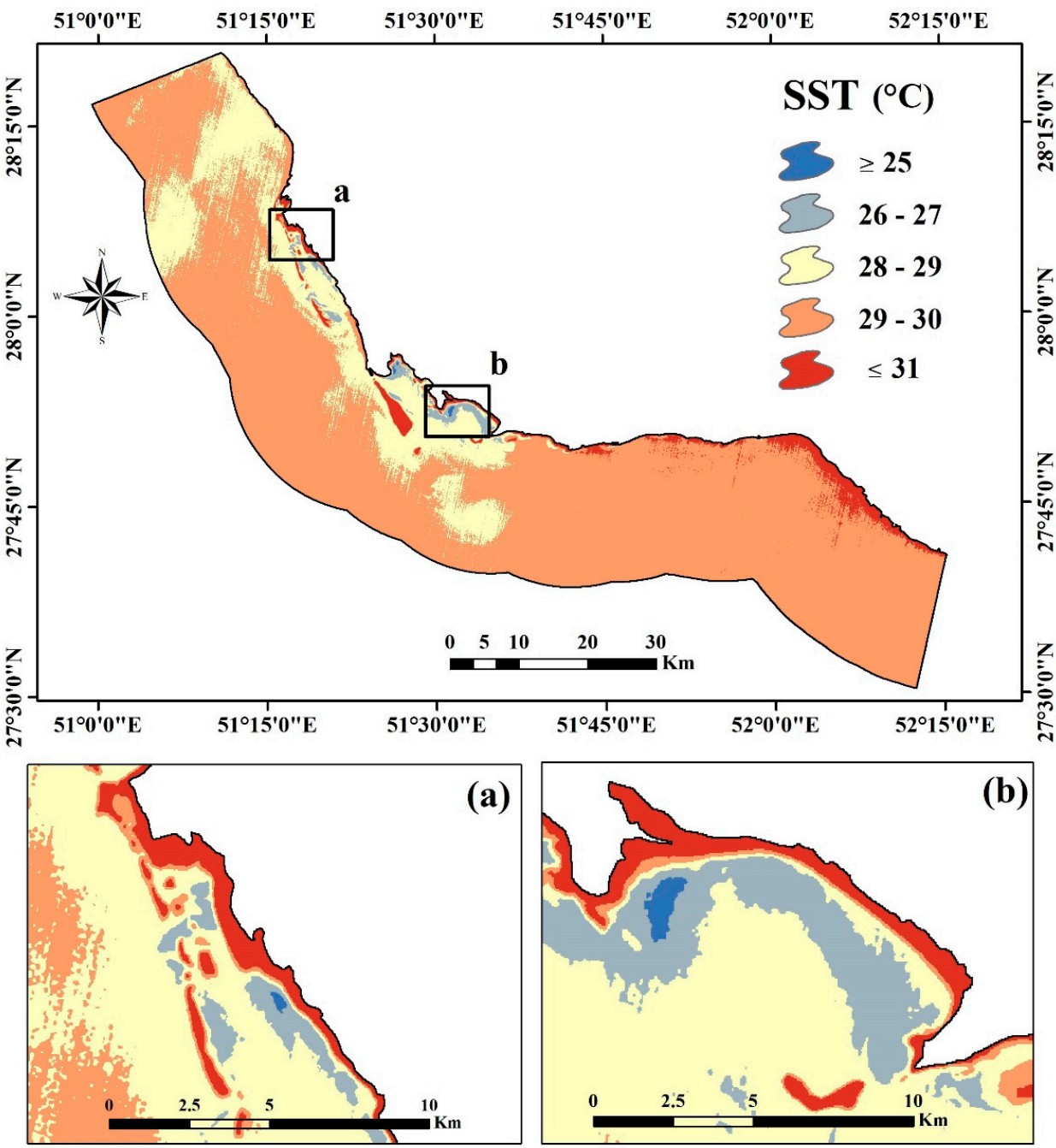

**Figure 3.** An example of the SST map derived from Landsat ETM+ TIR imagery acquired on 23 August 2015 (9:45 a.m. local time): (**a**) Shif-Hendijan and (**b**) Naiband #1.

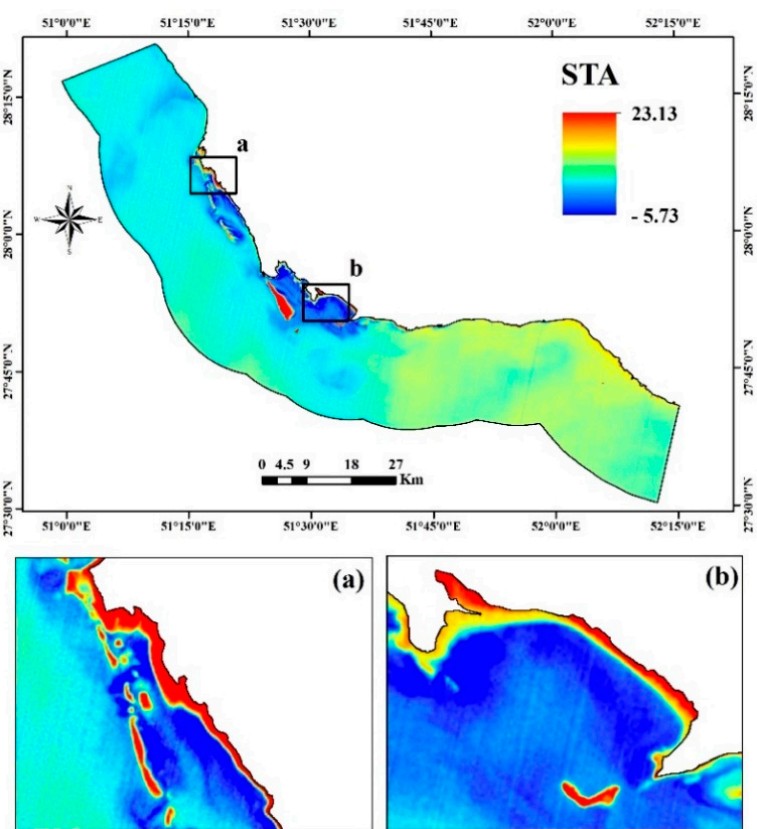

**Figure 4.** An example of the STA (standardized thermal anomaly) map of the study area on 23 August 2015 (9:45 a.m. local time): (**a**) Shif-Hendijan and (**b**) Naiband #1.

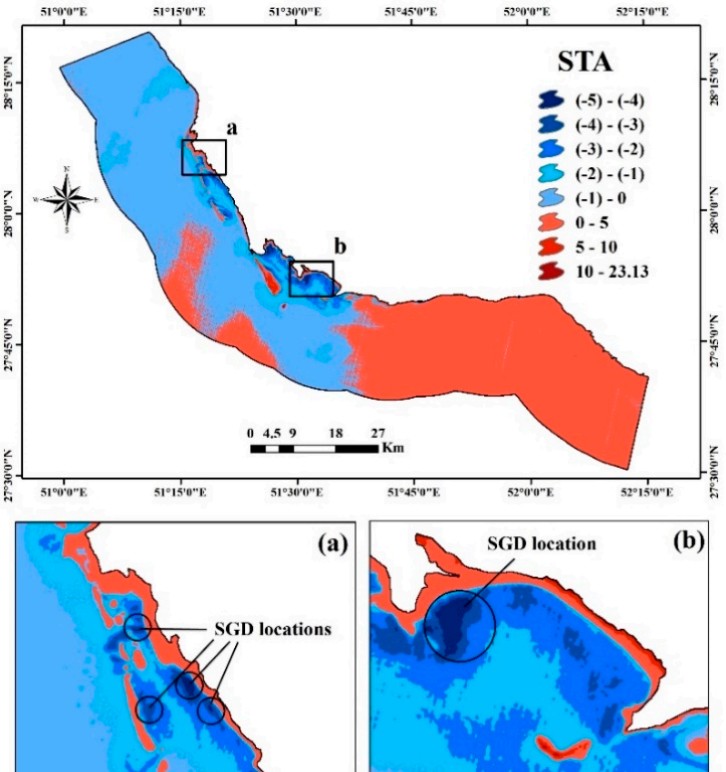

**Figure 5.** A reclassified STA map on 23 August 2015 and SGD location in the study area: (**a**) Shif-Hendijan and (**b**) Naiband #1. (negative temperature anomaly indicates SGD potential).

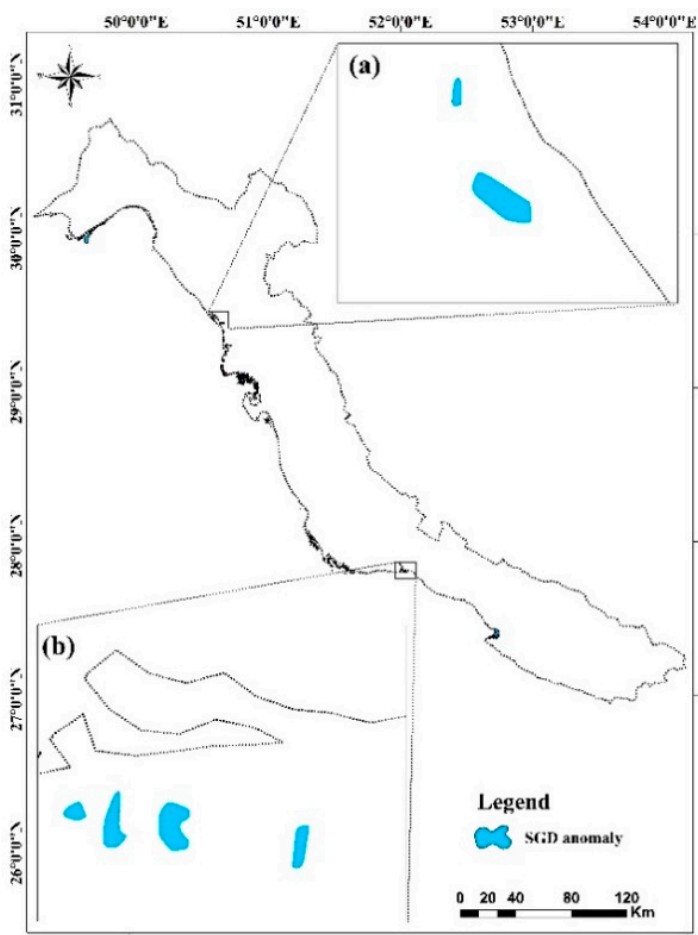

**Figure 6.** Geometric intersection of the anomalies based on all STA maps: (**a**) Shif-Hendijan and (**b**) Naiband #1.

**Table 1.** The area of thermal anomalies in 2015 and 2016 and their overlapping surface area for each Row/Pass pair. (The Row refers to the latitudinal center line of a frame of imagery while the Path is a line that the satellite moves along it).

| Row/Pass | Area of Thermal Anomaly in 2015 (ha) | Area of Thermal Anomaly in 2016 (ha) | Overlapping Surface Area in 2015 and 2016 (ha) |
|---|---|---|---|
| 162/41 | 2993 | 4566 | 2823 |
| 163/40 | 19,062 | 6956 | 6159 |
| 163/41 | 17,508 | 6916 | 4165 |
| 164/39 | 13,026 | 8518 | 4445 |
| 164/40 | 7752 | 5429 | 4725 |
| Total | 60,341 | 32,385 | 22,317 |

### 3.2. Statistical Comparison of SGD and Non-SGD Locations

The temperature of SGD and non-SGD locations in different parts of the study area was compared using the t-test (Table 2). The temperature of five sampling areas (20 sampling sites in five different areas) was measured. The spatial distance between samples (S1–S4) was greater than 1 km in each sampling area. It indicated that there are some differences in temperature values of each SGD and non-SGD. According to t-test results, significant differences were observed in terms of temperature in Naiband #1, Naiband #2, Bandargah, and Shif-Hendijan sites (Table 3). However, there is no significant differences between SGD and non-SGD in Dopalango-Khorkhan site in terms of temperature.

**Table 2.** In situ measurement of temperature in SGD and non-SGD sites.

| Sampling Areas | Temperature (in °C) | | | | | | | |
|---|---|---|---|---|---|---|---|---|
| | Sample 1 | | Sample 2 | | Sample 3 | | Sample 4 | |
| | SGD | Non-SGD | SGD | Non-SGD | SGD | Non-SGD | SGD | Non-SGD |
| Naiband #1 | 32 | 34.5 | 32.5 | 34.5 | 31 | 34.3 | 31.5 | 34.3 |
| Naiband #2 | 35 | 38.5 | 35 | 37.3 | 35.2 | 36.9 | 34.9 | 37.9 |
| Dopalango-Khorkhan | 28.5 | 30.5 | 28.6 | 30.2 | 28.8 | 29.5 | 27.5 | 30.6 |
| Bandargah | 28.6 | 29.7 | 28.4 | 29.7 | 27.5 | 29.8 | 26.8 | 29.8 |
| Shif- Hendijan | 23.5 | 25.5 | 23.5 | 25.5 | 23.5 | 25.5 | 23.5 | 25.5 |

**Table 3.** The *t*-test results of temperature between SGD and non-SGD sites.

| Parameter | Sampling Areas | | | | |
|---|---|---|---|---|---|
| | Naiband #1 | Naiband #2 | Dopalango-Khorkhan | Bandargah | Shif- Hendijan |
| Temperature | 0.032 * | 0.023 ** | 0.735 ns | 0.006 ** | 0.01 ** |

ns: not significant; ∗: $p < 0.05$; ∗∗: $p < 0.01$.

### 3.3. Relationships between SGD and Geo-Environmental Variables

Some of geo-environmental variables such as altitude, slope angle, TRI, VRM, surface ratio, spring density, rain index, temperature index, lineament density, and lineament intersection had a VIF value > 10 and TOL value < 0.1, consequently, these variables were excluded from the logistic regression analysis. The TOL value of other variables in this study was larger than 0.1, showing that there is no substantial multicollinearity between them.

After the forward stepwise logistic regression analysis, seven spring-affecting variables, which are the plan curvature (buffer 2), TPI (buffer 3), TWI (buffers 1 and 2), stream density (buffers 1 and 3), karstic area (buffers 2 and 3), NDVI (buffer 3), and aquifer area (buffer 1) were selected because they were statistically significant at the 95% level of confidence (Table 4). These variables, therefore, were taken to be influential predictor variables. However, some variables such as altitude, slope angle, TRI, VRM, surface ratio, profile curvature, spring density, rain index, temperature index, lineament density, and lineament intersection, that are generally accepted as groundwater-affecting variables were not found to be statistically significant in the model. Here, the coefficients (*b*) of all retained variables that are statistically different from zero have been estimated. According to the logistic regression, two types of correlation can be seen (Table 4). Some variables including plan curvature (buffer 2), TPI (buffer 3), TWI (buffers 1 and 2), percent of karstic area (buffers 2 and 3), NDVI (buffer 3), and aquifer area (buffer 1) had positive coefficients, while the logistic regression model shows a negative correlation between stream density (buffers 1 and 3) and SGDs.

Using the coefficients obtained from the final output of the logistic regression analysis, the form of logistic regression model can be shown as follows:

$$\begin{aligned}
Y = 1.544(Pc_2) \quad & + 1.435(TPI_3) + 3.927(TWI_1) + 11.389(TWI_2) \\
& - 18.793(SD_1) - 13.637(SD_3) + 21.2(PKA_2) + 43.2(PKA_3) \quad (11) \\
& + 1.29(NDVI_3) + 0.034(Aa_1) + 97.182
\end{aligned}$$

where *Pc* is plan curvature in buffer, *TPI* is *TPI* factor, *TWI* is the *TWI* factor, *SD* is stream density, *PKA* is the percent of karstic area and *Aa* is aquifer area in a given buffer area, the buffer defined by the sub-script.

**Table 4.** Results of logistic regression method.

| Variable | B [1] | S.E. [2] | Wald [3] | Sig. [4] |
|----------|-------|----------|----------|----------|
| $Pc_2$ | 1.544 | 6.72 | 0.052 | 0.021 |
| $TPI_3$ | 1.435 | 0.752 | 3.65 | 0.04 |
| $TWI_1$ | 3.927 | 1.658 | 5.61 | 0.018 |
| $TWI_2$ | 11.389 | 2.432 | 23.65 | 0.0 |
| $SD_1$ | −18.793 | 5.99 | 9.84 | 0.002 |
| $SD_3$ | −13.637 | 6.104 | 4.99 | 0.025 |
| $PKA_2$ | 21.2 | 2.523 | 70.60 | 0.009 |
| $PKA_3$ | 43.2 | 5.125 | 71.05 | 0.0 |
| $NDVI_3$ | 1.29 | 30.245 | 0.0018 | 0.0 |
| $Aa_1$ | 0.034 | 0.013 | 7.19 | 0.007 |
| Constant | 97.182 | 21.248 | 20.919 | 0.0 |

B [1] = logistic coefficient; S.E. [2] = standard error of estimate; Wald [3] = Wald chi-square values; Sig. [4] = significance.

### 3.4. Accuracy Assessment and Sensitivity Analysis

A total number of 326 pixels were identified as SGD. Of these pixels, 70% of the SGD inventory (228 pixels) were randomly selected for training the logistic regression model and the other 30% (98 pixels) were held as a validation data set. The area value under the ROC curve for the model was found to be 0.966 with an estimated standard error of 0.02 (Table 5). This result indicates that the model is an efficient estimator of the probability values of the SGD in the study area. As discussed in Yesilnacar [55], an AUC-ROC > 90% indicates an excellent predictive skill in the validation phase. Furthermore, since logistic regression model complexity is low, especially when there are no or few interaction terms [56], it is attractive for use in data-scarce regions.

**Table 5.** The result of validation step.

| Model | AUC Value | S.E. [1] | 95.0% C.I. for EXP(B) [2] | |
|-------|-----------|----------|--------|--------|
| | | | Lower | Higher |
| Logistic regression | 0.966 | 0.02 | 0.926 | 1 |

S.E. [1] = standard error of estimate; 95.0% C.I. for EXP(B) [2]: 95% confidence interval for Exp(B).

To investigate the contribution of independent variables to SGD modeling, a sensitivity analysis was undertaken. The results of sensitivity analysis (Table 6) indicate that all variables had a positive influence on the SGD prediction. The independent variables that appeared to have the most influence were the $PKA_3$ (RD = 17.81%), and $TWI_3$ (RD = 14.91%) (Figure 4). Some of the independent variables including $TWI_1$ (RD = 3.62%), $SD_1$ (RD = 3.42%), $PC_2$ (RD = 3.31%), $Aa_1$ (RD = 2.59%), $NDVI_3$ (RD = 1.66%) had a moderate contribution to SGD modeling. Conversely, a few of the variables contributed weakly to the modeling, notably $SD_3$ (RD = 0.52%) and $TPI_3$ (RD = 0.21%) (Figure 7). These results highlighted that the SGD occurrence is highly sensitive to the percent of karstic area and TWI. Subsurface karst consists of a range of caves and conduits, which provide complex pathways for groundwater.

**Table 6.** The result of sensitivity analysis.

| Excepted Factor | AUC Value | Accuracy (%) | 95.0% C.I. for EXP(B) [1] | |
|---|---|---|---|---|
| | | | Lower | Higher |
| $Pc_2$ | 0.934 | 93.4 | 0.870 | 0.999 |
| $TPI_3$ | 0.964 | 96.4 | 0.927 | 0.999 |
| $TWI_1$ | 0.931 | 93.1 | 0.868 | 0.994 |
| $TWI_2$ | 0.822 | 82.2 | 0.717 | 0.927 |
| $SD_1$ | 0.933 | 93.3 | 0.876 | 0.991 |
| $SD_3$ | 0.961 | 96.1 | 0.919 | 0.998 |
| $PKA_2$ | 0.911 | 91.1 | 0.839 | 0.983 |
| $PKA_3$ | 0.794 | 79.4 | 0.682 | 0.907 |
| $NDVI_3$ | 0.950 | 95.0 | 0.899 | 0.994 |
| $Aa_1$ | 0.941 | 94.1 | 0.883 | 0.999 |

95.0% C.I. for EXP(B) [1]: 95% confidence interval for Exp(B).

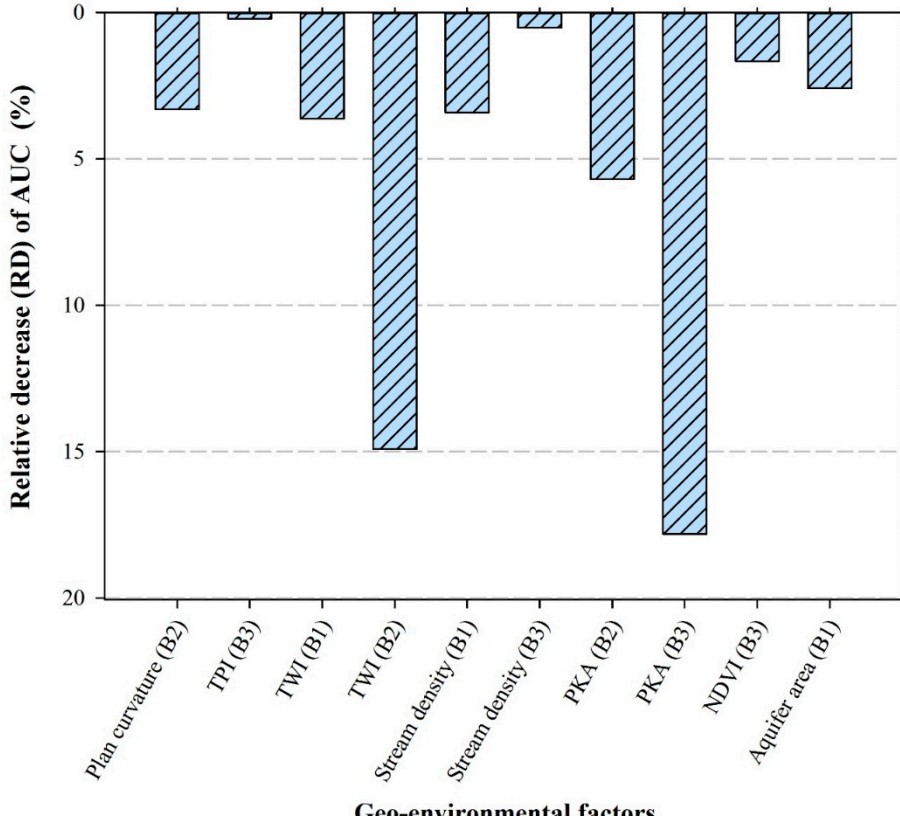

**Figure 7.** The relative decrease (RD) of AUC for each geo-environmental factor (B: buffer; TPI: topographical position index; TWI: topographic wetness index; PKA: percent of karstic area).

## 4. Discussion

### 4.1. Anomaly Mapping Using Thermal Remote Sensing

The applicability of Landsat 8 TIR data to identify SGD sites has been successfully used in previous studies [24,57]. This study also confirmed the capacity of thermal remote sensing for identifying SGD sites. Of course, it should be mentioned that as fresh water (i.e., groundwater discharge from coastal aquifers) is relatively buoyant compared to saline estuary waters, thermal signatures of groundwater discharge are easier to detect in estuaries compared with "fresh water–fresh water" interfaces (e.g., groundwater-lake interactions) where relatively cold water will not be detected immediately at the surface [57,58]. On the other hand, successful application of thermal analysis to identify sources of SGD is also constrained by the spatial resolution of the remote sensing system employed. However,

the spatial resolution of remote sensing images should be reasonable and needs a trade-off based on the both required precision and extent of the area. In this study, we worked on a wide region that application of drone or unmanned aircraft system (UAS) was limited. In fact, finer spatial resolution imagery could be obtained from the use of sensors mounted on drone or UAS approaches would likely serve to elucidate finer scale patterns of coastal water discharge and by doing so, highlight potentially numerous and significant SGDs on a local scale. The high spatial resolution Landsat 8 data has been known to be advantageous for applications in estuarine waters of coastal zones and. As discussed by Vanhellemont and Ruddick [59], imagery from Landsat 8 with an appropriate turbid water atmospheric correction, is thus of great practical to coastal water monitoring. From a temporal resolution, its revisit time is 16 days globally and SGD analysis in the summers can be conducted several times each year, especially in large region. However, as a limitation of Landsat 8, the images that were highly affected by clouds should be eliminated. In this study, Landsat thermal images were used to identify SGD and non-SGD sites. The temperature of five sampling areas (20 sampling sites in five different areas) was directly measured and statistically compared using the t-test. It indicated that there are some differences in temperature values of each SGD and non-SGD. According to t-test results, significant differences were observed in terms of temperature in Naiband #1, Naiband #2, Bandargah, and Shif-Hendijan sites. However, there is no significant differences between SGD and non-SGD in Dopalango-Khorkhan site in terms of temperature.

### 4.2. Relationships between SGD and Geo-Environmental Variables

The selection of variables influencing on SGD is one of the most important components in SGD assessment, and is helpful for developing models and designing field experiments [60]. In this regard, the results help to remove superfluous variables which result in saving money, time and effort by dropping unnecessary variables. In this study, to investigate the contribution of independent variables to SGD modeling, a sensitivity analysis was undertaken. The results of sensitivity analysis indicate that all variables had a positive influence on the SGD prediction. The independent variables that appeared to have the most influence were the $PKA_3$ (RD = 17.81%), and $TWI_3$ (RD = 14.91%). Some of the independent variables including $TWI_1$ (RD = 3.62%), $SD_1$ (RD = 3.42%), $PC_2$ (RD = 3.31%), $Aa_1$ (RD = 2.59%), $NDVI_3$ (RD = 1.66%) had a moderate contribution to SGD modeling. Conversely, a few of the variables contributed weakly to the modeling, notably $SD_3$ (RD = 0.52%) and $TPI_3$ (RD = 0.21%) (Figure 7). These results highlighted that the SGD occurrence is highly sensitive to the percent of karstic area and TWI. Subsurface karst consists of a range of caves and conduits, which provide complex pathways for groundwater. This is important result as karst features, generated by the dissolution of carbonate rocks such as limestone, are abundant with ~25% of the world's coastline [61]. In this regard, Einsiedl et al. [62], and Argamasilla et al. [63] demonstrated that the diversity of lithological units substantially conditions the groundwater interaction in coastal aquifers. Their results in relation to the role of lithological characteristics confirm our findings.

The results highlighted that the occurrences of SGD depend substantially on the percent of karstic area and topographic wetness index of the upstream zone which effect groundwater recharge, water-rock interaction processes (i.e., the residence time in the aquifer), the development of subsurface fractures, and the dissolution processes of karst geomorphology; which agree with the results of Mejías et al. [21] who investigated SGD from a karstic aquifer in the Western Mediterranean Sea (Castellón, Spain). From an environmental perspective, the results demonstrated that karstic geological formations significantly impact SGD occurrence and its hydrogeochemical processes, and hence the coastal and submarine environments [2,64]. Although pathways created by limestone dissolution in karst systems allow rapid infiltration and groundwater flow, there are some challenges in investigating karst aquifer because of karst development and high heterogeneity of subsurface flows within karst bedrock [65]. Additionally, the use of geomorphometric and topo-hydrological indices can facilitate the investigation of hydrological attributes

and processes governing the SGD from (upstream) terrestrial zones into the coastal lands. Whereas previously recorded in other parts of the world such as Turkey [66], Spain [21], Australia [67], USA [68], Taiwan [69], and Portugal [70] revealed that presence of SGD is related to upland karstic zones, they rarely established quantitative relationships and indices. Therefore, the findings may be generalized into the other similar regions for distinguishing the potential of SGD sites. Recently, Saleh et al. [71] assessed carbonate system in the Persian Gulf and demonstrated the variability of carbonate chemistry in the rocky intertidal shores. Geology of the upland areas controls water quality in some parts of the Persian Gulf. Plan curvature and aquifer area had a moderate contribution to the SGD occurrence. These variables are among the most important variables that controls the occurrence and movement of groundwater, especially in fractured bedrock aquifers [27]. However, some of predictive variables such as stream density, TPI and NDVI had a weak contribution to identifying SGD sites. A low value of variable importance implies that a predictor variable makes a weak contribution to the prediction process and the quality of the model output. The possible reason is that the relationships between SGDs and environmental variables are complicated and based on subsurface processes. Importantly, as discussed by Gevrey et al., 2003, investigating relative importance of predictive variables for geo-environmental models allows decision makers to design a more efficient conceptual model by selecting and ranking predictor variables. However, it should be noted that the relative importance of variables to a modeling process is considerably affected by the methods used [72] and a variable of limited importance in one model may be very important in another. Here, we only used a logistic regression model to assess the relative importance of variables in terms of the SGD occurrence. Further research, including the use of different types of model, should explore the issues in more detail. Importantly, as discussed by Gevrey et al. [73], investigating relative importance of predictive variables for geo-environmental models allows decision makers to design a more efficient conceptual model by selecting and ranking predictor variables.

## 5. Conclusions

SGD as a significant component of the water cycle is important in the management of coastal areas, mostly in arid and semi-arid regions where water scarcity is a serious issue. It is, however, difficult to study by traditional field-based research. The research highlighted the value of generating surface temperature maps from satellite sensor imagery to identify potential SGD interaction patterns. The following conclusions can be drawn from the results:

- The application of thermal images of Landsat in this study not only saved significant time and resources, but also was extremely effective. In addition, the study demonstrated that logistic regression showed an excellent performance in modeling the relations between the SGD occurrence and geo-environmental characteristics of the upstream area. According to field surveys and validation results, the approach used has allowed the accurate detection of coastal springs. The results will assist in understanding SGD formation and its spatiotemporal variation; as well as promote the development of strategies for the sustainable management of coastal and marine ecosystems. According to the results, evidently discernible cold-water plumes emanate from nearshore waters along Naiband, Asaloye, Dopalango, Dahane Tahmadan, Khorkhan, and Bandar Busher coastlines. In addition to the findings specific to the study area, the methodology may be transferable to other coastal regions with similar geological conditions.
- The sensitivity analysis indicated that the SGD is most sensitive to the PKA and TWI variables of the upstream area. Variables such as stream density, NDVI and TPI were the least important variables in the modelling SGD. Furthermore, the findings of this study could be useful for others such as ecologists, planners, and water resources managers in understanding how different aspects of geo-environmental variables

and the physicochemical mechanisms involved in groundwater recharge impact on SGD sources.

- The methodology can be applied to other similar regions as a rapid assessment of SGD occurrence. Future work should try to effectively manage upstream watersheds of this region because of their direct and indirect impacts on quantity and quality of SGDs. More research is needed and could usefully explore temporal variations of SGD as well as quantitative flux assessment.

**Author Contributions:** Data curation, M.F.; formal analysis, A.N.S.; funding acquisition, M.F.; investigation, A.N.S.; methodology, M.F., S.F., O.R. and G.A.K.; supervision, A.N.S.; validation, O.R.; visualization, O.R. and G.A.K.; writing—original draft, O.R. and M.F.; writing—review and editing, O.R., A.N.S., M.F., G.F. and A.M.M. All authors have read and agreed to the published version of the manuscript.

**Funding:** This research was funded by the Iran National Science Foundation (INSF) (Code No. 93034760).

**Data Availability Statement:** Publicly available datasets were analyzed in this study. This data can be found here: https://earthexplorer.usgs.gov/.

**Acknowledgments:** We thank the Iranian Department of Water Resources Management and Iranian Department of Geology for providing data and maps. We would like to thank the anonymous reviewers whose comments significantly improved the manuscript.

**Conflicts of Interest:** The authors declare no conflict of interest.

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
