# Peer review of "Scrutinizing Relationships between Submarine Groundwater Discharge and Upstream Areas Using Thermal Remote Sensing: A Case Study in the Northern Persian Gulf"

_remotesensing, doi:10.3390/rs13030358_

Round 1

Reviewer 1 Report

In this manuscript, the authors used Landsat 8 thermal sensor data to determine the potential sites of submarine groundwater discharge (SGD). Logistic regression modelling was used to identify the relationships between the remotely-sensed surface water temperature (SWT) patterns and geo-environmental variables of upland watersheds. The accuracy of the predictions was evaluated based on the AUC-ROC metric. The study area is the Persian Gulf coastline.

The results showed that the karst formations and topographic wetness index influence mainly SGD phenomenon and spatial distribution in the northern part of the study area.
The authors properly followed the correct methodology to validate the accuracy of the model with the area under the receiver operating characteristic curve (AUC-ROC) metric.
The area value under the ROC curve for the model was found to be 0.966 with an estimated standard error of 0.02 meaning that the model is an efficient estimator of the probability values of the SGD in the study area in 96.6% percentage.
The Presentation Quality and the Overall Merit of the manuscript are high while this methodology can be transferred to other coastal regions as a rapid assessment procedure for submarine groundwater discharge site detection.

Reviewer 2 Report

See my comments in the attached file.

Reviewer 3 Report

I find "Scrutinizing relationships between submarine groundwater discharge and upstream areas using thermal remote sensing and logistic regression model" to be adequate overall.  The concept of a regional approach remote sensing approach to minimize costly and time-consuming ground-based work is a great contribution of this article.  Grammatically, the manuscript reads well.  I found a few minor pluralization issues.  I strongly urge the authors to review the grammatical rules for using articles, in particular the definite article "the" and to a lesser extent the indefinite article "a" are missing from numerous sentences.  I have not fixed these issues in my attached detailed comments.  Generally, the introduction sets-up the manuscript nicely, the methods need more detail and clarification, the results and discussion are logical but could benefit from more detail, and the conclusions nicely summarize the main points of the manuscript.

Reviewer 4 Report

I welcome new data in a very poorly studied area.

The manuscript, however, is not ready for publication mainly because of the lack of controls. My primary concern is that the nearshore area can be low in temperature simply because of the upwelling of cold, subsurface water, or there may be some remnant winter water. For instance, Saleh et al.(Marine Pollution Bulletin, 2020, 151, Feb. 2020,110834) indicated that the same study area's temperature could be as low as 20C. As a result, some ground truthing is required. It is only when the salinity is low while the temperature is also low it can be concluded that the SGD is detected. BTW, Saleh et al. did catch some low T and low S coastal waters.

One minor point is that there are new developments relating the SGD to the coastal environment(e.g., Wang et al., Scientific Reports, 8, Article 11650,2018).

Round 2

Reviewer 2 Report

See my comments to the revised manuscript in the attached file.

Reviewer 4 Report

The authors do not seem to be able to provide ground-truthing data. As I mentioned before, the low-temperature  surface water, taken to be from the SGD, could actually come from the upwelling of deep water or the remnants of winter water. These possible sources of low-temperature surface water must be discussed.
